# Gradients in the mammalian cerebellar cortex enable Fourier-like transformation and improve storing capacity

Isabelle Straub[1†], Laurens Witter[1,2†], Abdelmoneim Eshra[1†], Miriam Hoidis[1†], Niklas Byczkowicz[1], Sebastian Maas[1], Igor Delvendahl[1‡], Kevin Dorgans[3§], Elise Savier[3#], Ingo Bechmann[4], Martin Krueger[4], Philippe Isope[3], Stefan Hallermann[1*]

[1]Carl-Ludwig-Institute for Physiology, Medical Faculty, Leipzig University, Leipzig, Germany; [2]Department of Integrative Neurophysiology, Center for Neurogenomics and Cognitive Research (CNCR), VU University, Amsterdam, Netherlands; [3]Institut des Neurosciences Cellulaires et Intégratives, CNRS, Université de Strasbourg, Strasbourg, France; [4]Institute of Anatomy, Medical Faculty, Leipzig University, Leipzig, Germany

**\*For correspondence:**
stefan_jens.hallermann@uni-leipzig.de

[†]These authors contributed equally to this work

**Present address:** [‡]Institute of Molecular Life Sciences, University of Zurich, Zurich, Switzerland; [§]Okinawa Institute of Science and Technology, Graduate University, Okinawa, Japan; [#]Biology and Psychology Department, University of Virginia, Charlottesville, United States

**Competing interests:** The authors declare that no competing interests exist.

**Abstract** Cerebellar granule cells (GCs) make up the majority of all neurons in the vertebrate brain, but heterogeneities among GCs and potential functional consequences are poorly understood. Here, we identified unexpected gradients in the biophysical properties of GCs in mice. GCs closer to the white matter (inner-zone GCs) had higher firing thresholds and could sustain firing with larger current inputs than GCs closer to the Purkinje cell layer (outer-zone GCs). Dynamic Clamp experiments showed that inner- and outer-zone GCs preferentially respond to high- and low-frequency mossy fiber inputs, respectively, enabling dispersion of the mossy fiber input into its frequency components as performed by a Fourier transformation. Furthermore, inner-zone GCs have faster axonal conduction velocity and elicit faster synaptic potentials in Purkinje cells. Neuronal network modeling revealed that these gradients improve spike-timing precision of Purkinje cells and decrease the number of GCs required to learn spike-sequences. Thus, our study uncovers biophysical gradients in the cerebellar cortex enabling a Fourier-like transformation of mossy fiber inputs.

## Introduction

Digital audio compression (e.g. 'MP3'; *Jayant et al., 1993*) and image compression (e.g. 'JPEG'; *Wallace, 1992*) rely on Fourier transformations, which decompose a signal (e.g. sound amplitude as a function of time or image intensity as a function of space) into its frequency components (power as a function of frequency). By storing these frequency components with different precision depending on psychophysical demands of hearing and seeing, the overall storage capacity can be increased dramatically. In principle, neuronal networks consisting of neurons with varied electrophysiological properties could be suitable for Fourier-like transformations of information. This could benefit processing in neuronal circuits by increasing the signal-to-noise ratio of input signals or by selecting only relevant spectral components of a signal. Interestingly, there are indications that for example pyramidal neurons in the visual cortex and in the hippocampus are tuned to different inputs or different input strengths (*Cembrowski and Spruston, 2019*; *Fletcher and Williams, 2019*; *Soltesz and Losonczy, 2018*). However, whether these neuronal networks perform a Fourier-like transform of their inputs remains unknown.

**eLife digest** The timing of movements such as posture, balance and speech are coordinated by a region of the brain called the cerebellum. Although this part of the brain is small, it contains a huge number of tiny nerve cells known as granule cells. These cells make up more than half the nerve cells in the human brain. But why there are so many is not well understood.

The cerebellum receives signals from sensory organs, such as the ears and eyes, which are passed on as electrical pulses from nerve to nerve until they reach the granule cells. These electrical pulses can have very different repetition rates, ranging from one pulse to a thousand pulses per second. Previous studies have suggested that granule cells are a uniform population that can detect specific patterns within these electrical pulses. However, this would require granule cells to identify patterns in signals that have a range of different repetition rates, which is difficult for individual nerve cells to do.

To investigate if granule cells are indeed a uniform population, Straub, Witter, Eshra, Hoidis et al. measured the electrical properties of granule cells from the cerebellum of mice. This revealed that granule cells have different electrical properties depending on how deep they are within the cerebellum. These differences enabled the granule cells to detect sensory signals that had specific repetition rates: signals that contained lots of repeats per second were relayed by granule cells in the lower layers of the cerebellum, while signals that contained fewer repeats were relayed by granule cells in the outer layers.

This ability to separate signals based on their rate of repetition is similar to how digital audio files are compressed into an MP3. Computer simulations suggested that having granule cells that can detect specific rates of repetition improves the storage capacity of the brain.

These findings further our understanding of how the cerebellum works and the cellular mechanisms that underlie how humans learn and memorize the timing of movement. This mechanism of separating signals to improve storage capacity may apply to other regions of the brain, such as the hippocampus, where differences between nerve cells have also recently been reported.

Controlling the timing and precision of movements is considered to be one of the main functions of the cerebellum. In the cerebellum, the firing frequency of Purkinje cells (PCs) (*Heiney et al., 2014*; *Herzfeld et al., 2015*; *Hewitt et al., 2011*; *Medina and Lisberger, 2007*; *Payne et al., 2019*; *Sarnaik and Raman, 2018*; *Witter et al., 2013*) or the timing of spikes (*Brown and Raman, 2018*; *Sarnaik and Raman, 2018*) have been shown to be closely related to movement. Indeed, cerebellar pathology impairs precision in motor learning tasks (*Gibo et al., 2013*; *Martin et al., 1996*) and timing of rhythmic learning tasks (*Keele and Ivry, 1990*). These functions are executed by a remarkably simple neuronal network architecture. Inputs from mossy fibers (MFs) are processed by GCs and transmitted via their parallel fiber (PF) axons to PCs, which provide the sole output from the cerebellar cortex. GCs represent the first stage in cerebellar processing and have been proposed to provide pattern separation and conversion of the MF input into a sparser representation (recently reviewed by *Cayco-Gajic and Silver, 2019*). These MF inputs show a wide variety of signaling frequencies, ranging from slow modulating activity to kilohertz bursts of activity (*Arenz et al., 2008*; *Rancz et al., 2007*; *Ritzau-Jost et al., 2014*; *van Kan et al., 1993*). Interestingly, in most cellular models of the cerebellum, each MF is considered to be either active or inactive with little consideration for this wide range of frequencies (*Albus, 1971*; *Marr, 1969*). Furthermore, in these models, GCs are generally considered as a uniform population of neurons.

Here, we show that the biophysical properties of GCs differ according to their vertical position in the GC layer. GCs located close to the white matter (inner-zone) preferentially transmit high-frequency MF inputs, have shorter action potentials, and a higher voltage threshold to fire an action potential compared with GCs close to the PC layer (outer-zone). These gradients in properties of GCs enable a Fourier-like transformation of the MF input, where inner-zone GCs convey the high-frequency, and outer-zone GCs the low-frequency components of the MF input. The different Fourier-like components are sent to PCs by specialized downstream signaling pathways, which differ in PF axon diameters, action potential conduction velocity, and PC excitatory postsynaptic potential

(EPSP) kinetics. Computational simulations show that the biophysical gradients in the GC and molecular layer significantly reduce the number of GCs required to learn a sequence of firing frequencies and reduce the time needed to switch between firing frequencies.

## Results

### Gradients in the biophysical properties of inner- to outer-zone GCs

To investigate whether GCs are tuned for different frequencies, we first investigated the intrinsic membrane properties of GCs from different depths within the GC layer in lobule V of the cerebellum of P21-30 mice. We divided the GC layer into three zones and performed whole-cell current-clamp recordings from inner- (closest to the white matter), middle- and outer-zone GCs (closest to PCs) (*Figure 1A,B*). Upon current injection, inner-zone GCs were less excitable compared with outer-zone GCs (*Figure 1C*). On average, the relationship between the frequency of action potentials and the injected current was surprisingly different for inner- and outer-zone GCs (*Figure 1D*): inner-zone GCs needed higher current injections to fire an action potential (inner: 56.8 ± 2.6 pA vs. middle: 51.2 ± 2.0 pA vs. outer: 39.4 ± 2.0 pA; n = 38, 25, and 37, respectively; $P_{Kruskal-Wallis}$ <0.0001; *Figure 1E*) and to achieve the maximum firing rate compared with middle- and outer-zone GCs (inner: 224.6 ± 9.8 pA vs. middle: 190.8 ± 9.6 pA vs. outer: 174.3 ± 9.0 pA, respectively; $P_{Kruskal-Wallis}$ = 0.002; *Figure 1F*). Consistently, inner-zone GCs had a more depolarized threshold for action potential generation compared with middle- and outer-zone GCs (−38.0 ± 0.7 mV vs. −38.2 ± 0.8 mV vs. −41.4 ± 0.6 mV; $P_{Kruskal-Wallis}$ = 0.003; *Figure 1G*) and a lower input resistance (486 ± 27 MΩ vs. 494 ± 27 MΩ vs. 791 ± 63 MΩ; $P_{Kruskal-Wallis}$ = <0.0001; *Figure 1H*). Furthermore, the capacitance of inner-zone GCs was significantly larger compared with outer-zone GCs (inner: 5.8 ± 0.2 pF vs. middle: 5.8 ± 0.2 pF vs. outer: 4.6 ± 0.1 pF; $P_{Kruskal-Wallis}$ = <0.0001 *Figure 1I*). In agreement with these findings, we observed depolarization block in inner-zone GCs at higher current inputs than for outer-zone GCs (*Figure 1C,D*). Furthermore, a larger delay of the first spike was observed in inner- compared with outer-zone GCs (*Figure 1J*; $P_{Kruskal-Wallis}$ = 0.0001; *Figure 1K*). The delay with 60 pA current injection was 48 ± 6 ms for inner-, 38 ± 4 ms for middle-, and 23 ± 2 ms for outer-zone GCs (n = 32, 25, and 37, respectively; note that 6 out of 38 inner-zone GCs did not fire an action potential at 60 pA). Finally, the action potential half-width of GCs differed significantly between the three zones (inner: 122 ± 2 μs vs. middle: 137 ± 4 μs vs. outer: 143 ± 4 μs; $P_{Kruskal-Wallis}$ = 0.0001; *Figure 1L*). The distribution of the raw data (*Figure 1—figure supplement 1*) suggests a gradual change in the average cell parameters along the depth axis of the GC layer, but two populations of neurons (*salt and pepper* distribution), or three populations of neurons (inner-, middle-, and outer-zone) cannot fully be ruled out.

To test whether these gradients are specific to lobule V, we investigated GCs in lobule IX. Here, we observed very similar gradients to lobule V (*Figure 1—figure supplement 2*). In short, outer-zone GCs were more excitable and had broader spikes compared with inner-zone GCs. Interestingly, the absolute values between lobule V and IX differed (*Figure 1—figure supplement 2*), consistent with previously described differences in, for example the firing frequency in vivo between these two lobules (*Witter and De Zeeuw, 2015a*; *Zhou et al., 2014*) and in the differential density of $K_v4$ and $Ca_v3$ channel expression in GCs across different lobules (*Heath et al., 2014*; *Rizwan et al., 2016*; *Serôdio and Rudy, 1998*). Taking the large functional difference between spino- and vestibulo-cerebellum into account (*Witter and De Zeeuw, 2015b*), these data suggest that different biophysical properties of GCs are likely a conserved mechanism throughout the entire cerebellar cortex, potentially tuning GCs to different frequencies.

Development can have large effects on the physiology of neurons, and GCs in particular undergo profound changes during development (*Dhar et al., 2018*; *Lackey et al., 2018*). To exclude confounding effects of the developmental stage, we tested whether these gradients were also present in more adult mice. Recordings obtained from GCs in lobule V in animals between 80 and 100 days of age revealed very similar gradients to those observed in young animals (*Figure 1—figure supplement 3*). Together, these data show prominent gradients in the electrophysiological properties of GCs over the depth of the GC layer, and that these gradients can consistently be found across different lobules and ages.

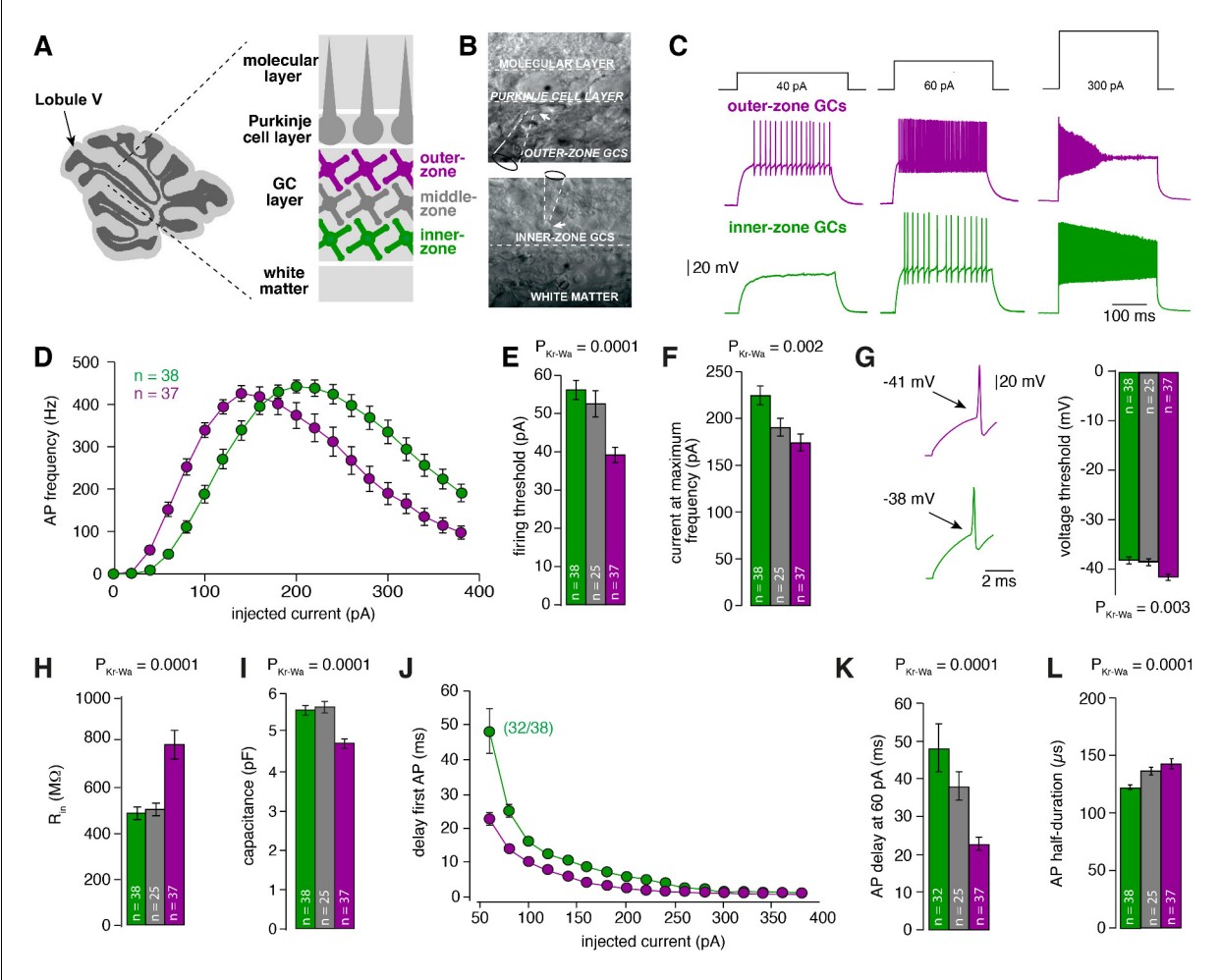

**Figure 1.** Gradients in the biophysical properties of inner- to outer-zone GCs. (**A**) Scheme of a parasagittal slice from the cerebellar cortex where lobule V is indicated by an arrow. Enlargement shows a schematic representation of the white matter, the GC, PC and molecular layer of the cerebellar cortex. Throughout the manuscript, inner-zone GCs (close to the white matter) are depicted in green, the middle-zone GCs in gray, and the outer-zone GC (close to the PCs) in magenta. (**B**) Example differential-interference-contrast (DIC) microscopic images of acute cerebellar slices during recordings from outer- (top) and an inner-zone GCs (bottom). The pipette is indicated with a dashed line. (**C**) Example current-clamp recordings from an outer-zone GC (magenta, top) and an inner-zone GC (green, bottom) after injection of increasing currents (40 pA, 60 pA and 300 pA). (**D**) Average action potential frequency from inner- (green, n = 38) and outer- (magenta, n = 37) zone GCs plotted against the injected current. Note that the maximum frequency is similar but outer-zone GCs achieved the maximum firing rate with a lower current injection (error bars represent SEM). (**E**) Average current threshold for action potential firing of inner-, middle- and outer-zone GCs ($P_{Dunns}$ = 0.0001 for inner- vs outer-zone GCs). All bar graphs represent mean and SEM. (**F**) Average current needed to elicit maximum firing frequency for inner-, middle- and outer-zone GCs ($P_{Dunns}$ = 0.001 for inner- vs outer-zone GCs). (**G**) Left: example action potentials from an inner- and outer-zone GC with the indicated (arrows) mean voltage-threshold for firing action potentials. Right: Comparison of the average voltage threshold for action potential firing ($P_{Dunns}$ = 0.006 for inner- vs outer-zone GCs). (**H**) Average input resistance of inner-, middle- and outer-zone GCs ($P_{Dunns}$ = 0.0001 for inner- vs outer-zone GCs). (**I**) Average capacitance of inner-, middle- and outer-zone GCs ($P_{Dunns}$ = 0.0001 for inner- vs outer-zone GCs). (**J**) Delay time of the first action potential plotted against injected current. Note that only 32 of 38 inner-zone GCs fired action potentials at a current injection of 60 pA. (**K**) Delay of the first action potential of inner-, middle- and outer-zone GCs at a current injection of 60 pA ($P_{Dunns}$ = 0.0001 for inner- vs outer-zone GCs). (**L**) Average action potential half-duration of inner-, middle- and outer-zone GCs ($P_{Dunns}$ = 0.0001 for inner- vs outer-zone GCs).

The online version of this article includes the following figure supplement(s) for figure 1:

**Figure supplement 1.** Raw data of the bar graphs from *Figure 1*.

**Figure supplement 2.** Gradients in the biophysical properties of GCs and PFs are preserved throughout the cerebellar cortex.

**Figure supplement 3.** Gradients in the biophysical properties of GCs and PFs are also found in 3-month-old animals.

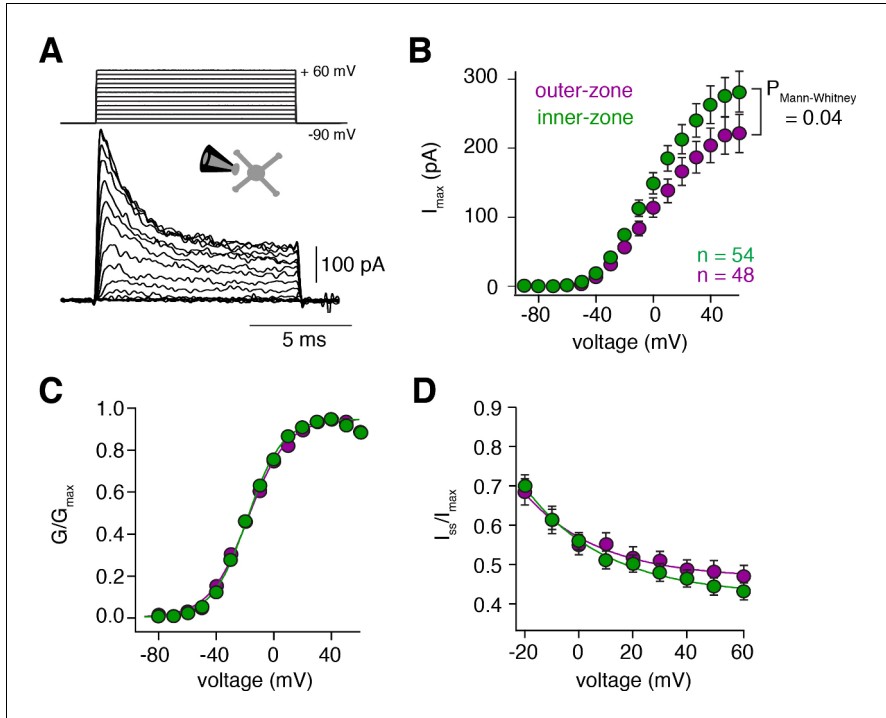

**Figure 2.** Voltage-gated potassium currents are larger at inner-zone GCs. (**A**) Example potassium currents from outside-out patches of cerebellar GCs evoked by voltage steps from −90 to +60 mV in 10 mV increments with a duration of 10 ms. All recordings were made in the presence of 1 μM TTX and 150 μM CdCl$_2$ to block voltage-gated sodium and calcium channels, respectively. (**B**) Average peak potassium current (I$_{max}$) plotted versus step potential of inner (green) and outer-zone (magenta) GCs. Significance level was tested with a Mann-Whitney *U* Test for the value at +60 mV and the p value is indicated in the figure. (**C**) Average normalized peak potassium conductance (G/G$_{max}$) versus step potential of inner (green) and outer-zone (magenta) GCs. (**D**) Average steady-state current (I$_{ss}$, mean current of the last 2 ms of the 10 ms depolarization) normalized to the peak current (I$_{max}$) versus step potential of inner (green) and outer-zone GCs (magenta).

The online version of this article includes the following figure supplement(s) for figure 2:

**Figure supplement 1.** Raw data of the amplitude of potassium currents at 60 pA current injection.
**Figure supplement 2.** Steady-state activation and inactivation are similar for inner and outer GCs.

## Voltage-gated potassium currents are larger at inner-zone GCs

To investigate possible biophysical causes for the gradients in the biophysical properties, we investigated voltage-gated potassium (K$_v$) currents by performing voltage-clamp recordings in outside-out patches from somata of inner- and outer-zone GCs in lobule V (*Figure 2A*). The maximum K$_v$ current at -60 mV was significantly higher in inner-zone GCs (282 ± 29 pA, n = 48) compared with outer-zone GCs (221 ± 28 pA, n = 54, P$_{Mann-Whitney}$ = 0.04; *Figure 2B*; *Figure 2—figure supplement 1*). Neither the steady-state activation curve (*Figure 2C*) nor the degree of inactivation (*Figure 2D*) was different between the two GC populations. Furthermore, steady-state inactivation, which was investigated with different holding potentials, was similar between inner- and outer-zone GCs (*Figure 2—figure supplement 2*). These data suggest that inner- and outer-zone GCs have a similar composition of K$_v$ channels, but inner-zone GCs have a higher K$_v$ channel density. The larger K$_v$ currents in inner-zone GCs are consistent with the short action potential duration of inner-zone GCs (cf. *Figure 1*). Thus, our data provide a biophysical explanation for the observed gradients in GC properties.

## MF inputs are differentially processed by inner- and outer-zone GCs

The gradients within the GC layer create an optimal range of input strengths for each GC. To test how these gradients impact the processing of synaptic MF inputs, we performed Dynamic Clamp experiments (*Desai et al., 2017*) and investigated whether different MF input frequencies

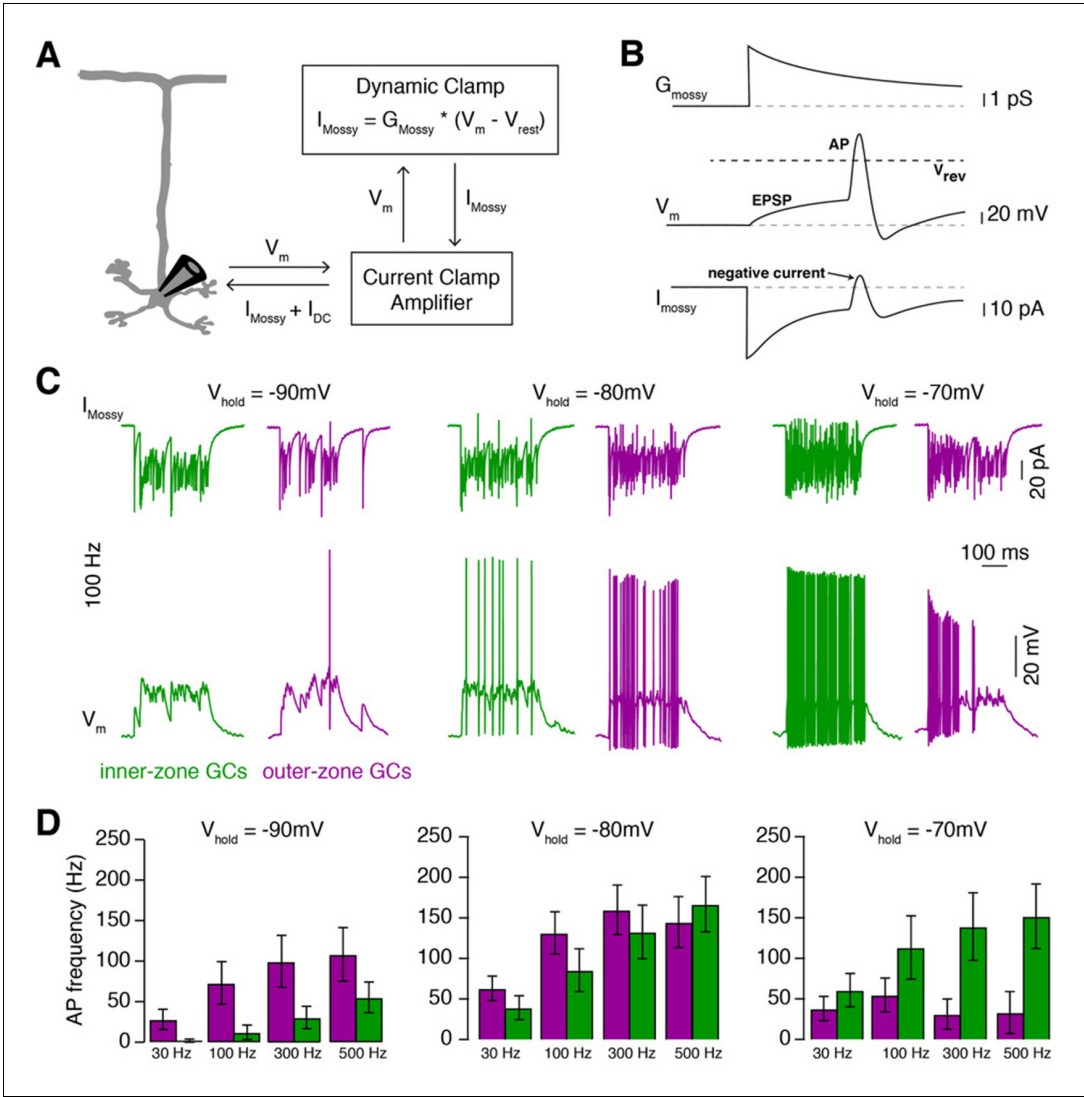

**Figure 3.** MF inputs are differentially processed by inner- and outer-zone GCs. (A) Schematic representation of the Dynamic Clamp system. (B) Illustration of MF conductance ($G_{mossy}$), GC membrane potential ($V_m$), and MF current ($I_{mossy}$) for the Dynamic Clamp technique. Note the prediction of a negative current during an action potential as apparent in the experimental traces in panel C. (C) Example Dynamic Clamp recordings of inner- (green) and outer-zone (magenta) GCs at different holding potentials (−90 mV left; −80 mV middle and −70 mV right) at a stimulation frequency of 100 Hz. Upper traces represent poisson-distributed MF currents. Lower traces show the measured corresponding membrane potential with EPSPs and action potentials. (D) Average firing frequency of inner- and outer-zone GCs during MF- like inputs at different frequencies and at the indicated holding potentials. The online version of this article includes the following figure supplement(s) for figure 3:

**Figure supplement 1.** MF input is similar for inner- and outer-zone GCs.
**Figure supplement 2.** Raw data of the bar graphs from *Figure 3*. Same data as in *Figure 3D*, but shown as box plots (median and interquartile range with whiskers) superimposed with single data points.

differentially affect spiking in inner- and outer-zone GCs (*Figure 3A and B*). We first recorded excitatory postsynaptic currents (EPSC) from GCs in inner- and outer-zones of lobule V after single MF stimulation. We found no significant differences in the amplitude or kinetics of EPSCs in inner- and outer-zone GCs (*Figure 3—figure supplement 1*).

Individual MFs span the entire depth of the GC layer, contacting both inner- and outer-zone GCs (*Krieger et al., 1985*; *Palay and Chan-Palay, 1974*). Furthermore, GCs are electronically extremely compact neurons and can be considered as a single compartment (*D'Angelo et al., 1993*;

*Delvendahl et al., 2015*; *Silver et al., 1992*). Therefore, we could use the Dynamic Clamp technique to implement the conductance of identical MF signals in inner- and outer-zone GCs based on the measured EPSC kinetics. We first applied input of a single MF with Poisson-distributed firing-frequencies ranging between 30 and 500 Hz for 300 ms duration while changing the resting membrane potential to simulate the large variability of membrane potential observed in GCs in vivo (*Chadderton et al., 2004*). In line with the gradients in the electrophysiological properties of GCs, inner-zone GCs fired fewer action potentials compared with outer-zone GCs in response to low-frequency MF inputs at a membrane potential of approximately –90 mV (*Figure 3C and D*; *Figure 3—figure supplement 2*). In contrast, inner-zone GCs fired more action potentials compared with outer-zone GCs in response to high-frequency MF inputs at a membrane potential of approximately –70 mV. In vivo, such a depolarization would be caused by reduced inhibition and/or simultaneous activation of multiple MF inputs. These data suggest that outer- and inner-zone GCs are specialized to process low- and high-frequency MF inputs, respectively.

## Fourier-like transformation of MF input frequency

To further test whether inner- and outer-zone GCs can extract different frequency components from a MF input signal, which would resemble a Fourier-transformation, we varied the MF input frequency sinusoidally between 30 and 300 Hz, representing a range of in-vivo-like tonic firing behaviour (*Figure 4A*; *Arenz et al., 2008*; *van Kan et al., 1993*). At a holding potential of −70 mV, commonly occurring in vivo (*Chadderton et al., 2004*), inner-zone GCs responded preferentially to high-frequency MF inputs up to 300 Hz, while outer-zone GCs responded preferentially to low-frequency inputs up to 100 Hz (*Figure 4B*; *Figure 4—figure supplement 1*). To estimate the optimal frequency at which inner- and outer-zone GCs preferentially fire action potentials, we calculated the phase angle (see Materials and methods, *Equation 3*). The mean phase angle, at which GC preferentially fired, was $162 \pm 8°$ for inner-zone (n = 10) and $100 \pm 20°$ for outer-zone GCs (n = 7; $P_{Mann-Whitney}$ = 0.02; *Figure 4C*), representing an average firing frequency of 284 and 116 Hz for inner- and outer-zone GCs, respectively (cf. *Equation 2*). Thus, the gradients in the biophysical properties enable the cerebellar GC layer to split incoming MF signals into different frequency bands and thereby to perform a Fourier-like transformation of the compound MF input signal.

## The position of PFs is correlated with the position of GC somata

A Fourier-like transformation in the GC layer (i.e. a separation of the spectral components of MF signals) could be particularly relevant if downstream pathways are specialized for these spectral components. Early silver-stainings and drawings from Ramón y Cajal indicate that inner-zone GCs give rise to PFs close to the PC layer and outer-zone GCs give rise to PFs close to the pia (*Eccles et al., 1967*; *Cajal, 1911* but see *Espinosa and Luo, 2008*; *Wilms and Häusser, 2015*). To test this possibility, we examined the axons of GCs. First, we investigated whether there is a correlation between the relative positions of the PF in the molecular layer and the GC somata in the GC layer. DiI was injected in vivo into the GC layer to label GCs and their axons. Several GCs were clearly stained 24 hr after DiI injection (*Figure 5A*), and the position of their soma and PF in the cerebellar cortex could be measured (*Figure 5B–D*). Even though the length of the ascending GC axon showed considerable variation ($196 \pm 5.5$ μm, range: 144 to 291 μm, n = 39 axons in n = 6 mice), after normalization for the thickness of the molecular and GC layers, the GC soma position was significantly correlated with the position of the bifurcation in the GC axon (*Figure 5C,D*; R = –0.86, p<0.001). These data show that inner-zone GCs preferentially give rise to PF located near the PC layer (inner-zone PFs) and outer-zone GC give rise to PF near the surface of the cerebellar cortex (outer-zone PFs).

## Inner-zone PFs have larger diameter and higher action potential conduction velocity

Next, we tested whether PFs, like GCs, have different properties depending on their position within the molecular layer. First, we compared the PF diameters in electron microscopic images of parasagittal sections of lobule V of mouse cerebellum and found significantly larger diameters for inner-zone PFs compared with middle- and outer-zone PFs ($182 \pm 2.6$ nm, n = 703 vs. $159 \pm 2.0$ nm, n = 819 vs. $145 \pm 1.7$ nm, n = 1024 *Figure 6A–C*; $P_{Kruskal-Wallis}$ <0.0001), which is in agreement with

previous investigations reported in cat (*Eccles et al., 1967*), monkey (*Fox and Barnard, 1957*), rat (*Pichitpornchai et al., 1994*), and mouse (*Wyatt et al., 2005*).

The axonal diameter is usually correlated with conduction velocity (*Jack et al., 1983*). We therefore recorded compound action potentials of PFs in lobule V and compared their conduction velocity in the inner-, middle-, and outer-zone of the molecular layer (*Figure 6D–F*). We detected a significantly higher velocity in inner-zone PFs compared with middle- or outer-zone PFs (0.334 ± 0.003 m*s$^{-1}$, n = 8 vs. 0.303 ± 0.004 m*s$^{-1}$, n = 6 vs. 0.287 ± 0.007 m*s$^{-1}$, n = 8; *Figure 6F*; P$_{Kruskal-Wallis}$ <0.0001). The absolute velocity and the gradient in the velocity from inner- to outer-zone PFs agree well with previous studies (*Baginskas et al., 2009*; *Vranesic et al., 1994*). These results suggest that inner-zone PFs are specialized for fast signaling, which is consistent with the concept that inner-zone GCs are tuned for high-frequency inputs (cf. *Figures 1* and *2*).

In addition to the above results obtained from lobule V, similar gradients in both axon diameter and axon conduction velocity were found in lobule IX (*Figure 6—figure supplement 1*). This suggests that gradients in axon diameter and axon conduction speed are general features of the cerebellar cortex.

A possible confounder of our results could be an over-representation of large-diameter Lugaro cell axons within inner-zone PFs (*Dieudonné and Dumoulin, 2000*). However, this would predict that the histograms of the axon diameters show two peaks with varying amplitude. Instead, we observed a single bell-shaped distribution in each zone (*Figure 6—figure supplement 2*), arguing that the

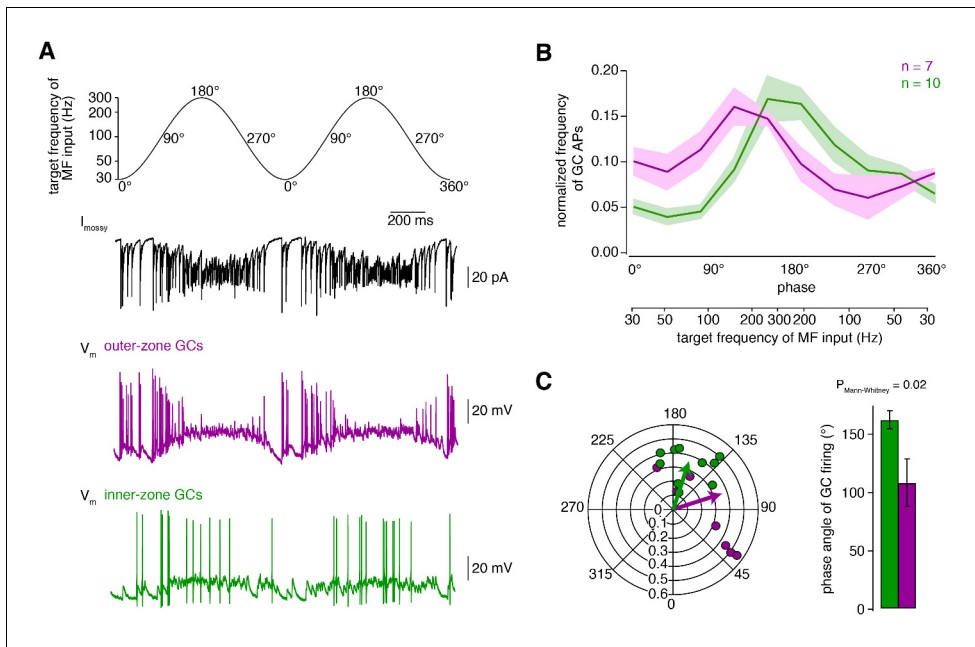

**Figure 4.** Fourier-like transformation of MF input frequency. (**A**) Target frequency of the Dynamic Clamp MF-like inputs during two cycles. The frequencies varied sinusoidally on a logarithmic scale between 30 to 300 Hz and the cycle duration was 1 s (*Equation 2*). The degree values denote the phase angle. *Black:* example trace of poisson-distributed MF-like inputs. *Magenta and green:* example membrane potential during Dynamic Clamp experiments of an outer- and an inner-zone GC, respectively, at a holding potential of approximately −70 mV. (**B**) Average normalized frequency of action potentials (APs) fired by inner- and outer-zone GCs (green and magenta, respectively) versus the phase angle and the target MF-like frequency within one cycle (for each cell, the integral of the spike histogram was normalized to 1; see *Figure 4—figure supplement 1* for absolute frequency). The light green and magenta areas represent the SEM. (**C**) Polar plot of phase angle and vector strength of the preferred firing frequency according to *Equation 3* from inner- (green) and outer-zone (magenta) GCs (*dots:* single cells; *arrows:* average). Bar graph of the average phase angle at which inner- and outer-zone GCs preferentially fired action potentials.

The online version of this article includes the following figure supplement(s) for figure 4:

**Figure supplement 1.** Raw data of the traces shown in *Figure 4B*.

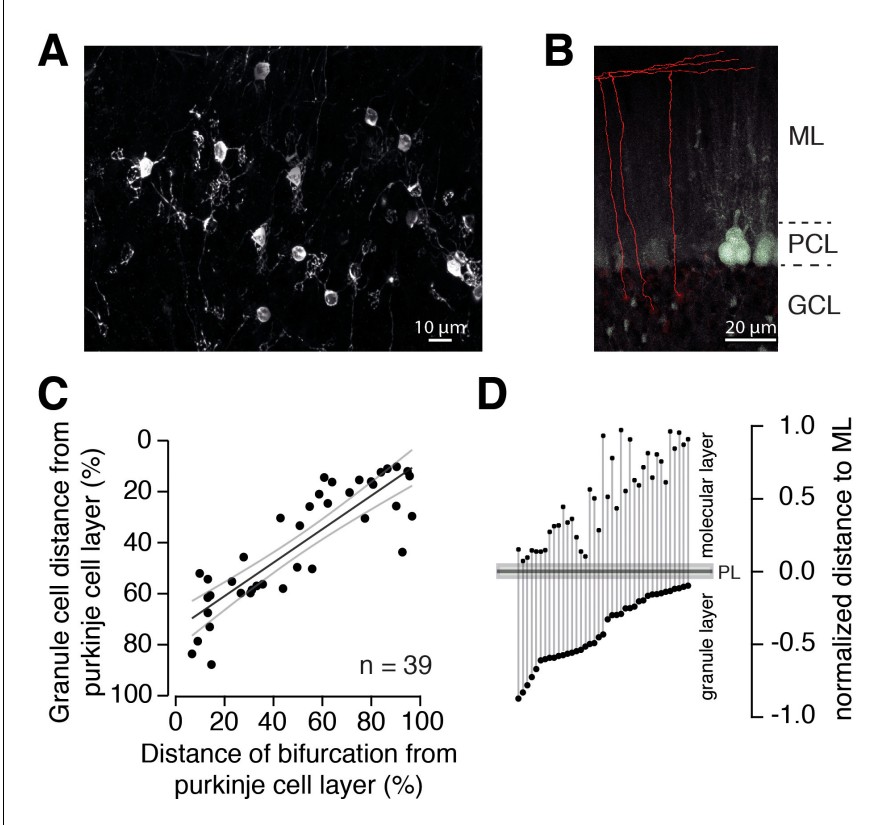

**Figure 5.** The position of PFs is correlated with the position of GC somata. (**A**) Example of GCs labeled with DiI 24 hr post injection. Numerous GCs from inner-, middle-, and outer-zone were labeled. (**B**) Example of traced axons from different GCs from the outer zone. The axon was traced (red) from the cell soma to the bifurcation site in the molecular layer. Stained cell bodies of GCs are also visible (white). *ML:* molecular layer; *PCL:* Purkinje cell layer; *GCL:* granule cell layer. (**C**) The distance between labeled GCs and the PC layer strongly correlated with the distance between the axon bifurcation and the PC layer (Pearson's correlation coefficient R = –0.86; p<0.001). Solid black line depicts the linear interpolation and the gray lines represent SEM of the fit. The number of GCs (n) is indicated. (**D**) Position of the GC somata within the GC layer of each traced cell linked to the position of the bifurcation site in the molecular layer. The distances were normalized to the height of the corresponding layers.

measured differences between axon diameters were not due to varying contributions from Lugaro cell axons, but reflect the differences between inner-, middle-, and outer-zone PFs.

## PCs process inner-, middle-, and outer-zone PF inputs differentially

Our data thus far indicate that GCs and PFs are adapted to different MF input frequencies and conduction velocities, respectively. This arrangement could in principle provide PFs with functionally segregated information streams that are differentially processed in PCs. To investigate this possibility, we made whole-cell current-clamp recordings from PCs in sagittal slices of the cerebellar vermis. PCs were held at a hyperpolarized voltage to prevent spiking and to isolate excitatory inputs. Electrical stimulation of PFs was performed at inner-, middle-, and outer-zones of the molecular layer and the stimulation intensity was adjusted to obtain similar EPSP amplitudes in all zones (*Figure 7A,B*). Stimulation of inner-zone PFs resulted in EPSPs (*Barbour, 1993*; *Roth and Häusser, 2001*) with shorter rise and decay times compared with EPSPs obtained from stimulating outer-zone PFs (rise$_{20-80}$: inner: 0.57 ± 0.04 ms, n = 12; middle: 0.93 ± 0.17 ms, n = 4; outer: 1.83 ± 0.33 ms, n = 12; P$_{Kruskal-Wallis}$ = 0.0001; decay: inner: 21.9 ± 1.5 ms, middle: 39.7 ± 1.1 ms outer: 40.8 ± 4.1 ms; P$_{Kruskal-Wallis}$ = 0.0004, *Figure 7C*; *Figure 7—figure supplement 1*). These results suggest that inner-zone PF inputs undergo less dendritic filtering in PCs compared with outer-zone PF inputs (*De Schutter and Bower, 1994a*; *Roth and Häusser, 2001* but see *De Schutter and Bower, 1994b*). To investigate high-frequency inputs to PCs, we elicited five EPSPs at 100 Hz and 500 Hz (*Figure 7D,E*).

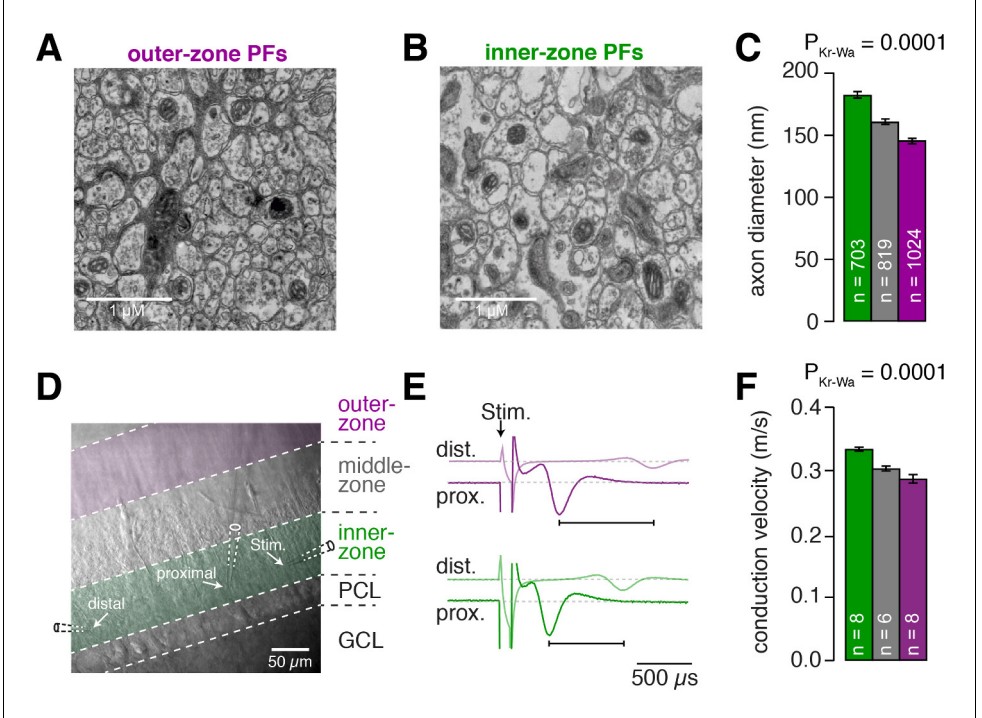

**Figure 6.** Inner-zone PFs have larger diameter and higher action potential conduction velocity. (**A**) Electron microscopic image of the outer (**A**) and inner zone (**B**) of sagittal sections through the molecular layer. (**C**) Summary of axon diameters in the inner- (green), middle- (gray), and outer-zone (magenta) of the molecular layer ($P_{Dunns}$ = 0.0001 for inner- vs outer-zone GCs). (**D**) DIC image of the molecular layer superimposed with a schematic illustration of the experimental setup to measure compound action potentials from PFs. Compound action potentials were evoked by a stimulus electrode (right) and recorded by a proximal and distal recording electrode (middle, left). (**E**) Example traces used to determine the conduction velocity of inner- and outer-zone PFs. The time difference between the compound action potential arriving at the proximal electrode (solid traces) and the distal electrode (light traces) was used to determine the velocity. The time was shorter for inner-zone PFs (green) compared with outer-zone PFs (magenta). (**F**) Summary of conduction velocity in inner-, middle- and outer-zones ($P_{Dunns}$ = 0.0007 for inner- vs outer-zone GCs).

The online version of this article includes the following figure supplement(s) for figure 6:

**Figure supplement 1.** Differences in axon diameter and conduction velocity are also found in lobule IX.

**Figure supplement 2.** Histogram of the axon diameters.

Individual EPSPs evoked from inner-zone PFs showed clear individual rising phases and peaks between each stimulus and less summation compared with outer-zone PFs (*Figure 7D–F*; *Figure 7—figure supplement 1*). These results suggest that inner-zone PFs can transmit timing information more faithfully compared with outer-zone PFs and thus control spike timing of PCs more precisely.

## The observed neuronal gradients increase storing capacity and improve temporal precision of PC spiking

Thus far we have described prominent gradients in the electrophysiological properties of GCs over the depth of the GC layer that enable inner- and outer-zone GCs to preferentially respond to high- and low-frequency inputs, respectively. The different frequency components are transferred via specialized PFs, which enable PCs to interpret high-frequency signals rapidly at the base of their dendritic trees and low-frequency signals slowly at more distal parts of their dendritic trees (*Figure 8A*).

To address the functional implications of these gradients in the GC and molecular layer, we performed computational modeling of a neuronal network of the cerebellar cortex with integrate-and-fire neurons. The model consisted of one PC and a varying number of GCs and MFs (*Figure 8A*). GCs received randomly determined MF inputs with either tonic (*Arenz et al., 2008*; *van Kan et al., 1993*) or bursting (*Rancz et al., 2007*) in-vivo-like spiking sequences. MF inputs were randomly

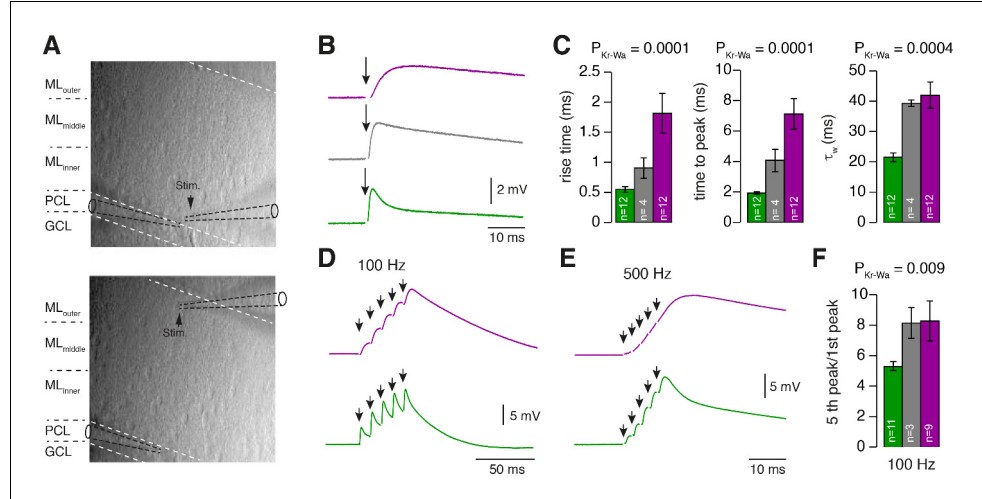

**Figure 7.** PCs differentially process inner-, middle-, and outer-zone PF inputs. (**A**) DIC image of the molecular layer superimposed with a schematic illustration of PC recordings while stimulating inner- (top) and outer-zone PFs (bottom). Shown are the GC layer (GCL), PC layer (PCL) and molecular layer (ML). (**B**) EPSPs measured at the PC soma after stimulation (1 Hz) of inner- (green), middle- (gray), and outer-zone PFs (magenta). (**C**) Average 20% to 80% rise time, time to peak and weighted decay time-constant of PC EPSPs after stimulation of inner- (green; n = 12), middle- (gray; n = 4) and outer-zone PFs (magenta; n = 12) as shown in B ($P_{Dunns}$ = 0.0001; $P_{Dunns}$ = 0.0001; $P_{Dunns}$ = 0.0009 for inner- vs outer-zone GCs, respectively). Note, one cell out of 12 had a monoexponentiell decay. (**D-E**) Example traces of EPSPs from a PC after five impulses to inner- (green) and outer-zone PFs (magenta) at 100 Hz (**D**) and 500 Hz (**E**). (**F**) Average paired-pulse ratio measured in PCs after five 100 Hz stimuli at inner- (green; n = 11), middle- (gray, n = 3) and outer- zone PFs (magenta, n = 8; $P_{Dunns}$ = 0.04 for inner- vs outer-zone GCs).

The online version of this article includes the following figure supplement(s) for figure 7:

**Figure supplement 1.** Raw data of the bar graphs from *Figure 7*.

distributed across layers, consistent with MFs having rosettes throughout the depth of the granule cell layer (*Krieger et al., 1985*; *Palay and Chan-Palay, 1974*). By changing the synaptic weights of the GC to PC synapses, the PC had to acquire a target spiking sequence with regular 80-, 40- and 120 Hz firing (*Figure 8B*). The algorithm for changing the synaptic weights was a combination of a learning algorithm based on climbing-fiber-like punishments and an unbiased minimization algorithm (see Materials and methods).

We first compared a model without gradients, where the parameters were set at the average of the experimentally determined values, with a model including all experimentally determined gradients (black and red, respectively, throughout *Figure 8*). To measure the difference between the final PC spiking and the target sequence, we calculated van Rossum errors using a time constant of 30 ms (*van Rossum, 2001*; *Figure 8C–E*). With an increasing number of GCs, the final PC spiking sequence resembled the target sequences increasingly better, as illustrated by an average spiking histogram from many repetitions with different random sets of MF inputs for models consisting of 100 and 1000 GCs (*Figure 8B*). As expected, the average minimal van Rossum error (for many repetitions with different random sets of MF inputs) decreased with increasing number of GCs (*Figure 8C*). For all sizes of the GC population, the average minimal van Rossum error was significantly smaller in the model containing all the experimentally determined gradients compared with the model without any gradients. For example, to obtain the spiking precision of the model containing 400 GCs with all gradients, the model without gradients required 800 GCs (cf. red arrows in *Figure 8C*). This indicates that for a cerebellum exploiting gradients, the number of GCs can at least be halved while obtaining a certain temporal precision compared with a cerebellum containing no gradients.

To investigate the relative contribution of each of the gradients, we tested models containing single gradients in isolation, resulting in intermediate van Rossum errors (blue, yellow, and green in *Figure 8C,D*). The average relative differences between the models across all sizes of the GC

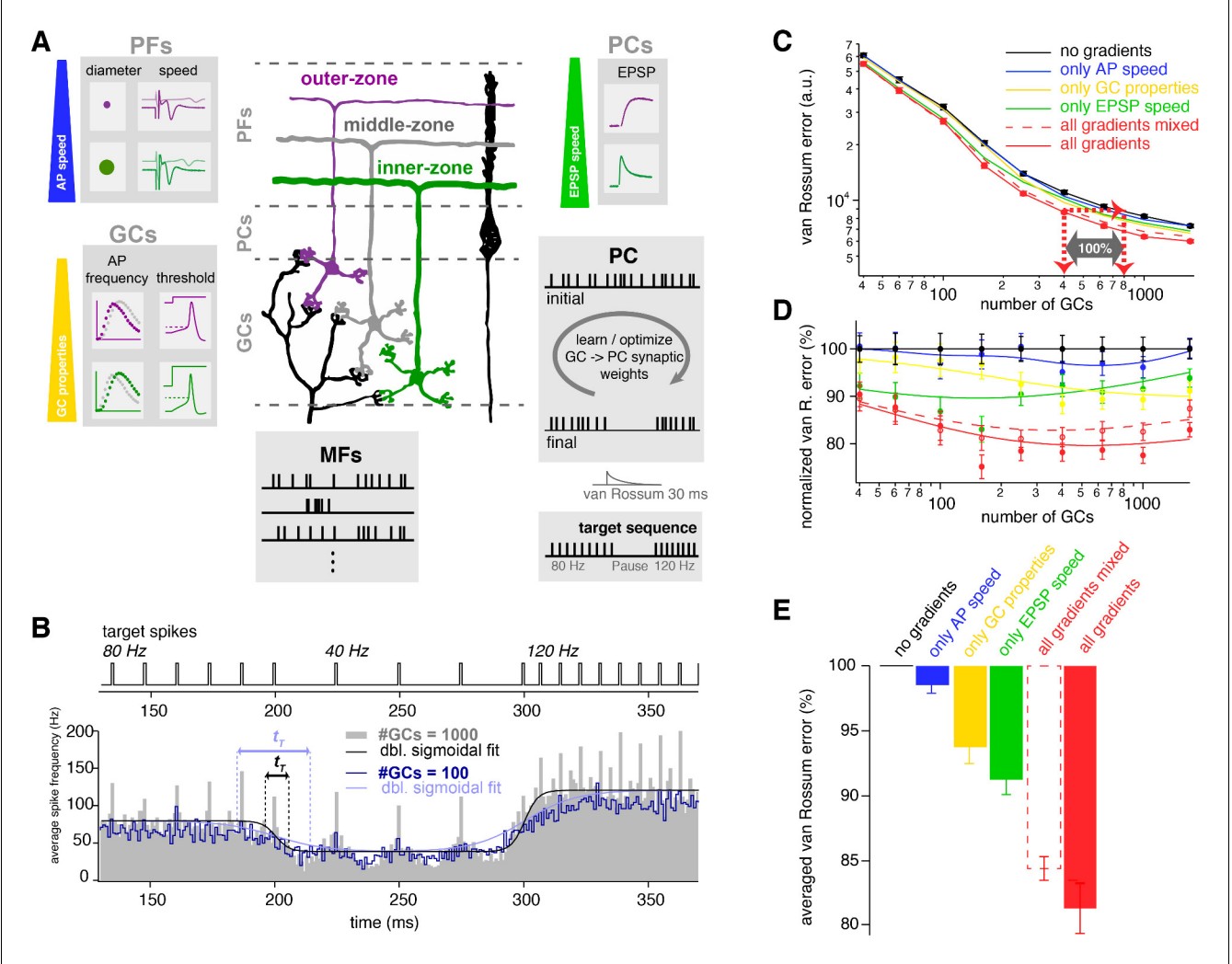

**Figure 8.** The observed neuronal gradients increase storing capacity and improve temporal precision of PC spiking. (**A**) Schematic illustration of the network model of the cerebellar cortex as explained in the main text. (**B**) Average spiking histogram for models consisting of 100 and 1000 GCs, superimposed with double sigmoidal fits constrained to 80, 40 and 120 Hz. The target spiking sequence is indicated above. $t_T$ indicates the transition time of the sigmoidal fit for the respective number of GCs. (**C**) Double logarithmic plot of the average minimal van Rossum error plotted against the number of GCs for models with no gradients (black), with only gradually varied PF conduction velocity (blue), GC parameters (yellow), )and EPSP kinetics (green), and with all gradients (red). Furthermore, all parameters were gradually varied but the connectivity between GC, PF and EPSPs was random (all gradients mixed; dashed red). Red dashed lines with arrows indicate the number of GCs needed to obtain the same van Rossum error with all gradients compared to no gradients. With no gradients, twice as many GCs are needed to obtain the same van Rossum error. (**D**) Average van Rossum error as shown in panel C but normalized to values obtaned from the model without gradients, superimposed with a smoothing spline interpolation. (**E**) Average of the relative differences shown in panel D.

The online version of this article includes the following figure supplement(s) for figure 8:

**Figure supplement 1.** The observed neuronal gradients reduce the temporal error and improve rate coding of PC spikes.

populations suggest an almost additive behavior of the individual gradients to the overall performance (*Figure 8E*).

To further investigate the interplay of the different gradients, we investigated a model containing all gradients, but the connectivity between GCs, PF action potential conduction velocity, and PC EPSP kinetics were randomly intermixed (red dashed lines in *Figure 8C–E*). The network benefits

from these intermixed gradients, but maximum optimization can only be obtained with correct connectivity (*Figure 8E*).

The time constant to calculate the van Rossum error can be decreased or increased to investigate spike timing or slower changes in firing rate, respectively. The impact of the gradients increased with increasing time constant (*Figure 8—figure supplement 1A,B*), indicating that rate coded signaling especially benefits from the here described gradients. To specifically test the effect of gradients on the cerebellum's ability to switch between firing frequencies, we made sigmoid fits around the times of firing rate changes. The transition time ($t_T$; see Materials and methods) from these fits showed that models with all gradients showed on average about 30% faster 'frequency-switching' than models without any gradients (*Figure 8—figure supplement 1C–F*).

Finally, we repeated the modeling experiments but used a target sequence with a firing pause (i.e. 80, 0, and 120 Hz instead of 80, 40, and 120 Hz) resulting in similar conclusions regarding the van Rossum errors and transition times (*Figure 8—figure supplement 1G–M*). A pause in firing enabled us to quantify the temporal error at the beginning and the end of the pause (*Figure 8—figure supplement 1N–Q*). These spike times have been proposed to be of particular relevance for behavior (*Hong et al., 2016*). Analysis of the temporal error in the beginning and the end of the pause revealed similar results compared with the van Rossum error and the transition time. Thus, our modeling results show that the experimentally determined gradients improve the spiking precision, accelerate 'frequency-switching', and increase the storing capacity of the cerebellar cortex.

## Discussion

In this study, we describe gradients in the biophysical properties of superficial to deep GCs, which enables the GC layer to perform a Fourier-like transformation of the MF input. Furthermore, we show that the downstream pathways from GCs to PCs are specialized for transmitting the frequency band for which the corresponding GCs are tuned to. Finally, computational modeling demonstrates that both the gradients in the GC layer and the specialized downstream pathways improve the spiking precision, accelerate the switching between firing frequencies of PCs, and increase storing capacity in the cerebellar cortex.

### Fourier-like transformation in the cerebellar cortex

Our data demonstrate that outer-zone GCs preferentially fire during MF input with low frequency ('low-frequency' GCs, magenta in *Figure 9A*), whereas inner-zone GCs preferentially fire during MF input with high frequency ('high-frequency' GCs, green in *Figure 9A*). The separation of a signal into its frequency components resembles a Fourier transformation (*Figure 9B*). The analogy with a Fourier transformation has the limitations that (1) the separation is only partial with overlapping ranges of preferred frequency, (2) a single MF cannot transmit two frequencies simultaneously but only separated in time (as illustrated in *Figure 9A*) and (3) concurrent inputs from two MFs with different frequencies synapsing onto a single GC cannot be separated. Yet, our data indicate that the entire GC layer with several MFs sending various frequencies to numerous GCs can execute a Fourier-like transformation. In analogy to the dispersion of white light into its spectral components by an optical prism, the broadband MF signal is separated into its spectral components with inner- to outer-zone GCs preferentially transmitting the high- to low-frequency components, respectively.

Such a partial separation offers the chance to differentially process high- and low-frequency components. Indeed, in the molecular layer, the high-frequency components of the MF signal are sent via rapidly conducting axons to proximal parts of the PC dendritic tree. This allows fast (phasic) signals to have a strong and rapid impact on PC firing. On the other hand, low-frequency components of the MF signal are conducted more slowly and elicit slower EPSPs, allowing slow (tonic) signals to have a modulatory impact on PC firing. Our data indicate that, in analogy to the increased storing capacity of digital audio and image compression (*Jayant et al., 1993*; *Wallace, 1992*), the combination of a Fourier-like transformation in the GC layer and specialized downstream signaling pathways in the molecular layer dramatically reduce the number of required GCs for precise PC spiking (*Figure 8*). Furthermore, our data support the 'adaptive filter' theory of the cerebellum, where broadband MF input is differentially filtered by GCs (*Dean et al., 2010*; *Fujita, 1982*; *Singla et al., 2017*). Within this framework, our data indicate gradients in the band-pass filtering properties of GCs. Furthermore, our data could provide an additional explanation for the improvement in motor learning

**Figure 9.** Illustration of the concept of Fourier-like transformation in the cerebellar cortex. (**A**) Illustration of a broadband MF input conveying a sequence of low, high, and low firing frequency. Inner-zone GCs will preferentially fire during high-frequency inputs ('high-frequency' GC) and outer-zone GCs during low-frequency inputs ('low-frequency' GC). (**B**) Schematic illustration of the signal flow through the cerebellar cortex. The Fourier-like transformation in the GC layer is illustrated as an optical prism separating the spectral components on the MF input. Thereby, the MF signal in the time domain is partially transformed into the frequency domain and sent to PCs via specialized signaling pathways in the molecular layer.

when elevating background activity of MFs (*Albergaria et al., 2018*): the elevated MF activity will help to overcome the high threshold of inner-zone GCs, which rapidly and effectively impact PCs via fast conducting PFs at the proximal dendrite.

## Axes of frequency specialization in the cerebellum

There are at least two axes of heterogeneity in the cerebellar cortex. First, Zebrin stripes can be observed as parasagittal zones ('medio-lateral' axis) in cerebellar cortex (*Apps et al., 2018*). Firing rate, firing regularity, synaptic connectivity and synaptic plasticity seems to differ between PCs in zebrin positive and negative zones (*Valera et al., 2016*; *Wadiche and Jahr, 2005*; *Xiao et al., 2014*; *Zhou et al., 2014*). Second, there is a lobular organization ('rostro-caudal' axis) as shown here by the functional differences between lobules V and IX (*Figure 1—figure supplement 1*). GCs in lobule IX are tuned to lower frequencies than GCs in lobule V. These findings are largely in line with previous investigations (*Heath et al., 2014*; *Witter and De Zeeuw, 2015a*; *Zhou et al., 2014*), where the anterior cerebellum was identified to process high-frequency or bursting signals, while the vestibulo-cerebellum mainly processed lower frequency or slowly-modulating inputs. Furthermore, the optimal time intervals for introduction of spike timing dependent plasticity differ between the vermis and the flocculus (*Suvrathan et al., 2016*).

In addition to these two known axes of heterogeneity, we described an axis that is orthogonal to the surface of the cerebellar cortex. This 'depth' axis causes inner-zone GCs to be tuned to higher frequencies than outer-zone GCs. The frequency gradients along the 'depth'-axes are in line with recently described connections of nucleo-cortical MFs and PC, which specifically target GCs close to the PC layer (*Gao et al., 2016*; *Guo et al., 2016*). These connections send slow feedback signals to the outer-zone GCs, which — according to our framework — are ideally suited to process such slow modulatory signals. Independent of these specialized feedback pathways, MFs exhibit heterogeneity (*Chabrol et al., 2015*; *Bengtsson and Jörntell, 2009*). Consistent with MFs having rosettes throughout the depth of the granule cell layer (*Krieger et al., 1985*; *Palay and Chan-Palay, 1974*), our data indicate that each type of the heterogeneous MF inputs is split into its frequency components along the depth axis. A preference of some MFs to specific zones could furthermore contributes to the frequency separation (*Quy et al., 2011*; *Jörntell and Ekerot, 2006*).

Our results predict that superficial GCs, such as the ones imaged recently in the investigation of eye-blink conditioning and reward representation in the cerebellar cortex (*Giovannucci et al., 2017*;

*Wagner et al., 2017*), would preferentially convey low-frequency signals to PCs and might not be representative for the full range of frequencies present over the depth of the GC layer. Recently, diverse adaptation of GCs to 2-s-lasting current injections has been described (*Masoli et al., 2019*), but it remains unknown to which extent this form of adaptation exhibits a gradient along the depth axis. The genetic reasons for the here-observed gradients in cerebellar cortex are currently not known. Due to a large variability within each zone, our data cannot rule out a *salt and pepper* distribution of two populations of neurons (*Espinosa and Luo, 2008*). However, neurons in the medial vestibular nucleus exhibit a graded tuning of the capacity for fast-spiking by expression levels of specific ion channels (*Kodama et al., 2020*). Thus, including this new 'depth' axis, there are three orthogonal axes along which the cerebellar cortex is tuned for preferred frequency, indicating the importance of proper frequency tuning of the circuitry.

## The role of inhibition

In the current study we did not investigate molecular layer interneurons, which can have a large impact on PC spiking (*Blot et al., 2016*; *Dizon and Khodakhah, 2011*; *Gaffield and Christie, 2017*; *Mittmann et al., 2005*; *Sudhakar et al., 2017*). However, the spatial arrangement of stellate and basket cell interneurons is consistent with our framework. Although the dendrites of molecular layer interneurons can span the entire molecular layer, the dendrites of basket cells seem to be preferentially located at the inner-zone of the molecular layer (*Palkovits et al., 1971*; *Rakic, 1972*), which positions them ideally to receive rapid high-frequency signals of inner-zone PFs. Consistently, they impact PC firing rapidly and efficiently via their pinceaus (*Blot and Barbour, 2014*). Furthermore, the dendrites of a subset of stellate cells (with their somata located in the outer-zone molecular layer) are preferentially located at the outer-zone molecular layer (*Palkovits et al., 1971*; *Rakic, 1972*), which positions them ideally to receive modulatory low-frequency signals and elicit slow IPSPs in PCs. Furthermore, molecular layer interneurons seem to represent a continuum along the vertical axis, with a correlation between the vertical location of the soma, axonal boutons, and dendrite location (*Sultan and Bower, 1998*), which is consistent with the here-described continuum of biophysical properties along the vertical axis of the cortex. Incorporating molecular layer interneurons, their synaptic plasticity and their potential gradients into the frequency-dispersion framework may show a further increase in the dynamic range of frequency separation within the cerebellar cortex c what we have described here (*Gao et al., 2012*).

## Functional implications for the cerebellum

MF firing frequencies range from <1 to ~1000 Hz (*Arenz et al., 2008*; *Chadderton et al., 2004*; *Jörntell and Ekerot, 2006*; *Rancz et al., 2007*; *van Kan et al., 1993*). Many previous modeling studies investigating cerebellar function considered the activity of each MF as a constant digital value (*Albus, 1971*; *Babadi and Sompolinsky, 2014*; *Brunel et al., 2004*; *Clopath et al., 2012*; *Marr, 1969*), a constant analog value (*Chabrol et al., 2015*; *Clopath and Brunel, 2013*), or spike sequences with constant frequency (*Billings et al., 2014*; *Cayco-Gajic et al., 2017*; *Steuber et al., 2007*). We focused on the time-varying aspects of MF integration in GCs, and therefore implemented a model with a corresponding large range of MF input frequencies that could change over time. It would be interesting to elucidate whether models with more uniform MF inputs, such as those found in many previous models, would benefit from the here-observed biophysical gradients.

To implement these gradients in a model, we used a simplified cerebellar circuitry that does not consider active dendrites (*Llinás and Sugimori, 1980*) or the tonic activity of PCs (*Raman and Bean, 1997*). It will therefore be interesting to investigate if the here-observed gradients in the GC and molecular layer improve the performance of more complex models of the cerebellar cortex (*De Schutter and Bower, 1994a*; *Garrido et al., 2013*; *Masoli et al., 2015*; *Medina et al., 2000*; *Rössert et al., 2015*; *Spanne and Jörntell, 2013*; *Steuber et al., 2007*; *Sudhakar et al., 2017*; *Walter and Khodakhah, 2009*; *Yamazaki and Tanaka, 2007*). Furthermore, it remains to be investigated whether gradients in the GC layer also improve models that aim to explain tasks such as eyeblink conditioning (*Mauk and Buonomano, 2004*) and vestibulo-ocular reflexes (*Lac et al., 1995*).

Our model simulated the learning that PCs undergo to acquire specific firing frequencies in response to GC input. PC firing rate and spiking precision have been shown to be closely related to movement (*Brown and Raman, 2018*; *Sarnaik and Raman, 2018*). Our results show that the same

temporal spiking precision or the same frequency switching speed can be obtained with approximately half the number of GCs when GC gradients are implemented (*Figure 8*). Taking into account the large number of cerebellar GCs in the brain (*Herculano-Houzel, 2009*; *Williams and Herrup, 1988*), a significant reduction in the number of GCs could represent an evolutionary advantage to minimize neuronal maintenance energy (*Howarth et al., 2012*; *Isler and van Schaik, 2006*). Therefore, the dramatic increase in storing capacity for precise PC spiking provides an evolutionary explanation for the emergence of gradients in the neuronal properties.

## Functional implications for other neural networks

Based on the described advantages of the Fourier transformation for rapid and storing-efficient information processing, we hypothesize that other neural networks also perform Fourier-like transformations and use segregated frequency-specific signaling pathways. To our knowledge, this has rarely been shown explicitly, but similar mechanisms might operate, for example, in the spinal cord network: descending motor commands from the pyramidal tract send broadband signals to motoneurons with different input resistances resulting from differences in size. This enables small motoneurons to fire during low-frequency inputs and large motoneurons only during high-frequency inputs (*Henneman et al., 1965*). Furthermore, specialized efferent down-stream signaling pathways innervate specific types of muscles with specialized short-term plasticity of the corresponding neuromuscular junctions (*Wang and Brehm, 2017*).

In the hippocampus, frequency preferences of hippocampal neurons are well established in enabling segregation of compound oscillatory input into distinct frequency components (*Pike et al., 2000*). Furthermore, there is increasing evidence that what has been considered a homogeneous population of neurons exhibit gradients in the neuronal properties (*Cembrowski and Spruston, 2019*), such as the intrinsic electrical properties and synaptic connectivity in CA3 pyramidal neurons (*Galliano et al., 2013*). The here reported heterogeneity furthermore enables functional segregation of information streams for example in CA1 pyramidal neurons (*Soltesz and Losonczy, 2018*). Additionally, gradients in biophysical properties of neurons in the entorhinal cortex might serve to generate functional outcomes relevant for the generation of grid cell sizes (*Giocomo et al., 2007*; *Schmidt-Hieber and Nolan, 2017*; *Orchard et al., 2013*). Finally, in the neocortex, gradients in anatomical and biophysical properties were recently uncovered (*Fletcher and Williams, 2019*).

In summary, our findings contribute to the growing body of evidence that the neurons of a cell layer can exhibit systematic functional heterogeneities with differential tuning of neurons along gradients. Our data furthermore suggest that such gradients facilitate complex transformation of information, such as Fourier-like transformations, to cope with a broad temporal diversity of signals in the central nervous system.

# Materials and methods

## Electrophysiology

Parasagittal 300-µm-thick cerebellar slices were prepared from P21–P30 (young animals) or from P80– P100 (old animals) C57BL/6 mice of either sex as described previously (*Ritzau-Jost et al., 2014*; *Delvendahl et al., 2015*). Animals were treated in accordance with the German and French Protection of Animals Act and with the guidelines for the welfare of experimental animals issued by the European Communities Council Directive. The extracellular solution for the whole-cell measurements contained (in mM): NaCl 125, NaHCO$_3$ 25, glucose 20, KCl 2.5, CaCl$_2$ 2, NaH$_2$PO$_4$ 1.25, MgCl$_2$1 (310 mOsm, pH 7.3 when bubbled with Carbogen (95%O$_2$/5%CO$_2$)). For outside-out measurements of potassium currents (*Figure 2*), 150 µM CdCl$_2$ and 1 µM TTX were added to the external solution to block voltage-gated calcium channels and sodium channels, respectively. The intracellular solution contained in mM: K-Gluconate 150, NaCl 10, K-Hepes 10, Mg-ATP 3, Na-GTP 0.3, EGTA 0.05 (305 mOsm, pH 7.3). A liquid junction potential of +13 mV was corrected for. All electrophysiological measurements were performed with a HEKA EPC10 amplifier (HEKA Elektronik, Lambrecht/ Pfalz, Germany) under control of the Patchmaster software. All measurements were performed at 34–37˚C.

## Current clamp recordings in GCs

Action potentials were evoked in current-clamp mode by current pulses (amplitude 20–400 pA, duration 300 ms). To determine the input resistance, subthreshold current pulses were applied from −20 to + 20 pA in 2 pA steps. The resistance of the solution-filled patch-pipettes was between 6 and 12 MΩ . Data were sampled at 200 kHz.

## Outside-out recordings in GCs

To reliably clamp potassium currents from the soma of GCs (*Figure 2*), potassium currents were measured in outside-out patches pulled from the soma of inner and outer GCs by applying 10-ms voltage steps from −90 to +60 mV with 10 mV increments at an intersweep interval of 1 s. The intersweep holding potential was −90 mV. Data were sampled at 100 kHz.

## Compound action potentials in PFs

For the detection of compound action potentials in PFs, two pipettes (tip resistances 1–4 MΩ) filled with extracellular solution and connected to the patch-clamp amplifier were positioned within the molecular layer of horizontally cut slices of the cerebellar vermis. The average distance between two recording electrodes was 143 ± 5 µm. Compound action potentials were evoked by voltage stimulation (100 V) for 100 µs with a third pipette connected to an accumulator powered stimulation device (ISO-Pulser ISOP1, AD-Elektronik, Buchenbach, Germany). 40 to 80 stimulations delivered at 1 Hz were averaged and analyzed.

## Excitatory postsynaptic potentials in PC

Excitatory postsynaptic potentials (EPSPs) in PC were elicited by voltage stimulation of the PFs within the inner, middle or outer third of the molecular layer from horizontally cut cerebellar slices (*Figure 7*). 10 µM SR95531 was added to the external solution to block GABA$_A$ receptors. The stimulation pipette was filled with extracellular solution, and the voltage was adjusted between 6 to 25 V to elicit EPSPs with amplitudes between 1 and 2 mV. EPSPs were measured after a single 100 µs voltage stimulation or five stimulations (100 µs duration) at a frequency of 100 and 500 Hz. Averages of 30 trains per stimulation protocol were used for data analysis.

## Excitatory postsynaptic currents in GCs

To measure evoked EPSCs from GCs (*Figure 3—figure supplement 1*), 90–100 days-old mice were used. GCs from inner- or outer-zone from lobule V were held at resting conditions and MF axons were stimulated at 1 Hz with a second pipette. The average stimulation voltage was 36 ± 3 V for outer-zone GCs and 37 ± 3 V for inner-zone GCs.

## **Dynamic Clamp of MF conductance in GCs**

In order to analyze the response of GCs on in vivo-like MF inputs, we used a Dynamic Clamp implemented with the microcontroller Teensy 3.6 (https://www.pjrc.com) as described by *Desai et al. (2017)*. The Teensy was programmed using the Arduino integrated development environment with the code provided by *Desai et al. (2017)* and modified for our need as described in the following.

The time course of MF conductance was

$$G_{EPSC}(t) = G_{max} A_{norm} \left( -e^{-\frac{t}{\tau_r}} + \sum_{i=1}^{3} a_i e^{-\frac{t}{\tau_i}} \right) \qquad (1)$$

where the exponential rise time ($\tau_r$) was 0.1 ms, the decay time constants ($\tau_1$, $\tau_2$, and  $\tau_3$) were 0.3, 8, and 40 ms, respectively, and the relative amplitude of the decay components ($a_1$, $a_2$, and $a_3$) were 0.7, 0.26, and 0.04, respectively. The peak conductance ($G_{max}$) was 1 nS (*Hallermann et al., 2010*) and the normalization factor ($A_{norm}$) was 0.518, which was numerically calculated to obtain a peak amplitude of 1. The kinetics of the MF conductance were chosen to reproduce the measured mixed AMPA and NMDA EPSC kinetics of single EPSCs (*Figure 3—figure supplement 1*) and trains of EPSCs (*Baade et al., 2016*). The short-term plasticity during Poisson sequence of spikes was implemented by changing $G_{max}$ according to a simple phenomenological model (*Tsodyks and Markram,*

*1997*) assuming a release probability $p_{r0}$ of 0.4 (*Ritzau-Jost et al., 2014*). Facilitation was implemented as an increase in the release probability according to $p_r = p_r + 0.2*(1- p_r)$ and decaying back to $p_{r0}$ with a time constant of 12 ms (*Saviane and Silver, 2006*). Depression was implemented according to a recovery process with a time constant of 25 ms, which approximates a biexponential recovery process of 12 ms and 2 s (*Hallermann et al., 2010*; *Saviane and Silver, 2006*). The resulting short-term plasticity reproduced previously obtained data with regular spiking ranging from 20 to 1000 Hz (*Baade et al., 2016*; *Hallermann et al., 2010*; *Ritzau-Jost et al., 2014*).

The microcontroller was programmed to implement the MF conductance and its short-term plasticity with Poisson distributed spike times with a constant frequency ranging from 30 to 500 Hz for 300 ms (*Figure 3*). In each cell, each frequency was applied five times.

To investigate the response to sinusoidally varying input frequencies (*Figure 4*), the target frequency of the Poisson process ($F$) was varied on a logarithmic scale according to:

$$F(t) = exp(\log(F_{min}) + (\log(F_{max}) - \log(F_{min}))(0.5 - 0.5\ cos(2\pi t/T))) \tag{2}$$

where the minimal and maximal frequency ($F_{min}$ and $F_{max}$) were 30 and 300 Hz, respectively, and the duration of the sine wave cycle ($T$) was 1 s. In each cell, 10 cycles were applied consecutively for at least four times (interval >30 s). The histogram of the spike times (*Figure 4B*) was averaged across the last four cycles of all cells. The vector strength and phase angle (*van Kan et al., 1993*) were calculated as the absolute value and the argument of the complex number $\rho$ ($i = \sqrt{-1}$):

$$\rho = \frac{1}{N}\sum_{n=1}^{N} e^{i2\pi\frac{t_n}{T}} \tag{3}$$

where $t_n$ are the spike times of all $N$ spikes per experiment and $T$ the cycle duration (1 s). To increase statistical validity, only those cells that fired more than 100 action potentials during the analyzed cycles were included in the analysis. This criterion resulted in the exclusion of 3 out of 13 and 2 out of 9 cells for inner- and outer-zone GCs, respectively. However, inclusion of these cells in the analysis resulted in similar preference for MF firing frequency [phase angle along the cycle: $146 \pm 10°$ for inner-zone (n = 13) and $103 \pm 18°$ for outer-zone GCs (n = 9; $P_{Mann-Whitney} = 0.06$), representing an average firing frequency of 246 and 123 Hz for inner- and outer-zone GCs, respectively].

## Electron microscopy

Four C57BL/6 mice of either sex with an age between P23–P28 were sacrificed, followed by transcardial perfusion with saline and consecutively a fixative containing 4% paraformaldehyde and 2% glutaraldehyde in phosphate-buffered saline (PBS). After removal of the brain, the tissue was allowed to post-fix over night at 4°C and sagittal sections of the cerebellum were prepared at a thickness of 60 μm using a Leica microtome (Leica Microsystems, Wetzlar, Germany). The sections were stained in 0.5% osmium tetroxide in PBS for 30 min followed by dehydration in graded alcohol and another staining step with 1% uranyl acetate in 70% ethanol. After further dehydration, the tissue was embedded in durcupan (Sigma-Aldrich), which was allowed to polymerize for 48 hr at 56°C between coated microscope slides and cover glasses. Regions of interest were identified by light microscopy, cut and transferred onto blocks of durcupan to obtain ultra-thin sections using an Ultramicrotome (Leica Microsystems). Ultra-thin sections were transferred onto formvar-coated copper grids and stained with lead citrate. Ultrastructural analysis was performed using a Zeiss SIGMA electron microscope (Zeiss NTS, Oberkochen, Germany) equipped with a STEM detector and ATLAS software.

## Measurement of parallel-fiber axon diameter

Electron micrographs were manually analyzed in a blind manner (numbered by masked randomization) and each micrograph was divided into eight identically sized fields. The diameter of each parallel-fiber axon was measured as the longest chord in one or two of these fields. Cross sections with visible active zones or mitochondria were excluded from analysis.

## Dil injections and GC tracking

Six P20 CD1 mice were anesthetized with isoflurane (4%). An incision of the skin to expose the skull was made and a hole was manually drilled using a 25G needle above the desired injection site.

Injections of small amounts of DiI (1,1-dioctadecyl-3,3,3,3 tetramethylindocarbocyanine perchlorate, ThermoFisher Scientific, 10% in N,N-dimethylformamide) were performed using a broken glass pipette connected to a picospritzer II (Parker Instrumentation). 24 hr after injection, animals were sacrificed and transcardially perfused with 4% paraformaldehyde in PBS. The cerebellum was dissected, fixed overnight, and embedded in 4% agarose in PBS. 150 µm thick sections were then cut in the transverse or sagittal plane using a vibratome (VT1000, Leica microsystems). Z-Stacks (1 µm steps) were acquired using a confocal microscope (Leica SP5 II, 63x objective). GCs were traced from their soma to the axonal bifurcation of PFs. (Average stack depth: 84 ± 20 µm). GC axons were reconstructed using the 'Simple Neurite Tracer' plugin (*Longair et al., 2011*) in Fiji (ImageJ, NIH, USA). This plugin allowed us to assess the continuity of axons between several cross-sections. GC ascending axons were then fully traced and measured within the Z-limits of image sections. The size of the different layers of cerebellar cortex was reconstructed in each Z-stack. To avoid variability, all distances were normalized to the corresponding molecular layer height.

## Data analysis

Current-clamp data were analyzed using custom-written procedures in Igor Pro software (WaveMetrics, Oregon, USA) as previously described (*Eshra et al., 2019*). Intrinsic properties of GCs were determined from the injected currents that elicited the largest number of action potentials. The action potential threshold was defined as the membrane voltage at which the first derivative exceeded 100 V s$^{-1}$, the minimal action potential peak was set at −20 mV and the minimal amplitude at 20 mV. All action potentials with a half-width shorter than 50 µs and longer than 500 µs were excluded. Action potential voltage threshold and half-width were calculated from the average of the first five action potentials. If a trace contained less than five action potentials, only the first action potential was considered. The action potential frequency was determined by dividing the number of action potentials during the 300-ms-lasting current injection by 300 ms. Membrane capacitance, resting membrane potential and series resistance were read from the amplifier software (HEKA) after achieving the whole-cell configuration. Input resistance ($R_{in}$) was analyzed from alternating subthreshold current injections from −20 to 20 pA (2 pA steps). The resulting voltage was plotted against the injected current and a spline interpolation was performed to obtain the slope at the holding membrane potential (0 pA current injection).

The peak-current from outside-out patches was determined from voltage steps (−90 to +60 mV) with Fitmaster software (HEKA). Steady-state inactivation was determined from the last 2 ms of the respective sweep. Cells were only included if 50 pA <$I_{max}$ < 1 nA to exclude potential whole-cell measurements and membrane-vesicles.

EPSP measurements from PCs and EPSC measurements from GCs were analyzed with the Fitmaster software (HEKA). For PC EPSPs, 20–80% rise time and time to peak were determined from the average of 30 individual single EPSPs. GCs EPSCs were averaged from 25 traces. To obtain the decay kinetics, single EPSPs/EPSCs were fitted with either one or two exponentials. The weighted time constant was calculated as:

$$\tau_{w} = \frac{A_{slow}\tau_{slow} + A_{fast}\tau_{fast}}{A_{slow} + A_{fast}} \tag{4}$$

Paired-pulse ratio was determined between the first and the 5$^{th}$ EPSP after stimulation with 100 Hz trains. Single EPSCs from inner- and outer-zone GCs were averaged and fitted with two exponentials. The decay kinetics and amplitude of the grand-average was used to implement the MF EPSCs for the Dynamic Clamp.

## Neuronal networking modeling

The neuronal network consisted of varying numbers of MF inputs, GCs, and one PC and was implemented in Matlab (The MathWorks, Inc, Natick, Massachusetts, R2017a). For each simulation, a random set of MF inputs was generated. This input was then fed to a layer of integrate-and-fire GCs. An integrate-and-fire PC received the output of the GCs as EPSPs with delays based on PF conduction velocity. The PF-to-PC synaptic weights were optimized with the aim to make the PC spiking sequence similar to the target sequence. In the following, each component of the model is explained in detail.

## MF inputs

To simulate in vivo-like MF firing patterns, half of the MFs fired tonically (*van Kan et al., 1993*) and the other half fired bursts (*Rancz et al., 2007*). All MF spike trains were modeled first by generating a 'threshold trace'. For tonically firing MFs, this threshold trace was a Gaussian function with a peak and standard deviation chosen from uniform distributions ranging between 10 and 100 Hz and 0.2 and 0.5 s respectively, and a peak time point between 0 and 0.5 s. For burst firing MFs, the threshold trace was an exponential function with a peak randomly chosen between 600 and 1200 Hz, a decay time constant of 30 ms and a peak between 0 and 0.5 s. The threshold trace was then evaluated against random numbers from a uniform distribution to determine the occurrence of a spike. To accelerate the simulations, the sampling time interval was adjusted to 1 ms.

## GC properties

GCs were implemented as integrate and fire models with the following parameters: membrane resistance linearly varied between 450 MΩ for inner GCs to 800 MΩ for outer GCs (*Figure 1H*) and the threshold linearly varied between −37 mV for inner GCs to −42 mV for outer GCs (*Figure 1G*). For the models without the GC gradient, these values were set to the mean of the values for the inner and outer GC (i.e. 625 MΩ and −39 mV). The reset potential was set to −90 mV and the membrane potential to −80 mV.

## PF properties

To simulate a different action potential conduction velocity, the GC spike times were delayed by a value linearly varied between 0 for inner and 3 ms for outer GCs. The delay was calculated as the difference in conduction time required to travel 5 mm with a speed of 0.28 and 0.33 m s$^{-1}$ (*Figure 4F*). Even with this anatomically rather too large PF length (*Harvey and Napper, 1991*), the PF conduction speed had only a small impact on the model performance (see e.g. blue lines and bars in *Figure 7C–E*), arguing against a big impact of PF conduction delays (*Braitenberg et al., 1997*) at least in our model approach.

## Synaptic connections and properties

Each MF was connected to 10 GCs and each GC received 2 MF inputs, that is, the number of MF was 1/5 of the number of GCs. Since our model consists only of 'active' MFs, we chose only 2 and not 4 MFs per GCs (*Billings et al., 2014*). The MF to GC synapse was implemented as a model with one pool of vesicles with a release probability of 0.5 and a vesicle recruitment time constant of 13 ms (*Hallermann et al., 2010*). Synaptic facilitation was implemented by increasing the release probability after each spike by 0.2 decaying to the resting release probability with a time constant of 12 ms (*Saviane and Silver, 2006*). The synaptic conductance had exponential rise and decay time constants of 0.1 and 2 ms, respectively, and a peak amplitude of 1.9 nS (*Silver et al., 1992*). Correspondingly, the GC to PC synapse was implemented as a model with one pool of vesicles with a release probability ($p_{r0}$) of 0.4 and a vesicle recruitment time constant of 50 ms. Synaptic facilitation was implemented by increasing the release probability after each spike by 0.2 decaying to the resting release probability with a time constant of 50 ms (*Doussau et al., 2017*; *Isope and Barbour, 2002*; *Valera et al., 2012*). The synaptic conductance had an exponential rise time constant between 0.5 and 2 ms and a decay time constant between 17.5 and 70 ms for inner- and outer-zone GCs, respectively (*Figure 6*). The peak amplitude was adjusted to equalize the charge of the EPSC and to generate an approximately correct number of PC spikes (with the initial start values, that is, all GC to PC synaptic weight factors = 1) by linearly varying between 0.5 and 0.15 nS for inner- and outer-zone GCs, respectively.

## PC properties

The PC was implemented as an integrate and fire model with a membrane resistance of 15 MΩ, resting membrane potential of −50 mV, and a firing threshold of −45 mV. Spontaneous firing of PCs (*Raman and Bean, 1997*) was not implemented, and the only inputs to drive PCs to threshold were the GC-to-PC EPSCs.

## Target sequence and van Rossum error

Based on in vivo firing patterns (*Witter and De Zeeuw, 2015a*), an arbitrary target firing sequence of 80, 40, and 120 Hz for 300, 100, and 100 ms, respectively, was chosen. The distance between the PC and the target spiking sequence was quantified with the van Rossum error (*van Rossum, 2001*). Both spiking sequences were convolved with an exponential kernel with a decay time constant of 30 ms (or values ranging from 2 to 300 ms in (*Figure 8—figure supplement 1A,B*). The van Rossum error was defined as the integral of the square of the difference between these two convolved traces. We also tested another algorithm to calculate the van Rossum error (*Houghton and Kreuz, 2012*), C++ code taken from http://pymuvr.readthedocs.io/ and incorporated into Matlab via the MEX function and results were comparable.

## Learning and minimization algorithm

For each random set of MF inputs, the GC to PC synaptic weights were changed according to the following algorithm with the aim to minimize the van Rossum error between the PC spiking sequence and the target sequence. The initial values of the synaptic weights were 1, and values were allowed to change between 0 and 100. First, an algorithm was used that was based on supervised learning (*Raymond and Medina, 2018*) to punish the GCs that have spikes that precede unwanted PC spikes. Subsequently, an unbiased optimization of the GC to PC synaptic weight was performed using the patternsearch() algorithm of Matlab to minimize the van Rossum error. To increase the chance that a global (and not local) minimum was found, the minimization of the routine was repeated several times with random starting values. Other optimization routines such as a simplex [(fminsearch() of Matlab) or a genetic algorithm (ga() of Matlab)] revealed similar results. To exclude the possibility that the differences in the minimal van Rossum error between models with and without gradients were due to a bias in our learning algorithm, we performed a set of simulations with networks consisting of less than 100 GCs, in which we skipped the learning algorithm and only used an unbiased minimization algorithm. This resulted in similar difference in the minimal van Rossum error between models with and without gradients, indicating that the learning algorithm was not biased toward one type of model. For networks consisting of more than 100 GCs the prelearning was required to facilitate the finding of the global minimum.

300 different sets of random MF inputs were used to determine 300 statistically independent minimal van Rossum values for each of the models with a different number of GCs and a different number of implemented gradients (illustrated as mean ± SEM in *Figure 8C*). Comparing different models with the same set of MF input (using the nonparametric paired Wilcoxon signed-rank statistical test) the difference was significant (p<0.001) for all of the models and all number of GCs. The van Rossum errors were then normalized to the mean of the error of the model without gradients (*Figure 8D*). The values in *Figure 8D* were fitted with cubic spline interpolation using the logarithm of the number of GCs as abscissa.

To quantify the transition time between two target frequencies of the PC, the spike histogram was fitted with the equation

$$f(t) = 80 + \frac{-80 + 40}{1 + e^{-(t-200)/t_T}} + \frac{-40 + 120}{1 + e^{-(t-300)/t_T}} \tag{5}$$

where $f$ is the spike frequency in Hz and $t$ the time in ms. The transition time $t_T$ corresponds to the 23% to 77% decay and rise time for the transition from 80 to 40 Hz and from 40 to 120 Hz, respectively.

## Sensitivity of model parameters

We verified that our conclusions do not critically depend on specific parameters of the model. For example, decreasing the simulation time interval from 1 ms to 100 μs, resulted in a difference of the best van Rossum error of 21% between models with and without gradients consisting of 100 GC, compared with a difference of 17% between the corresponding models with the default simulation time interval of 1 ms (cf. *Figure 8D*). With 4 MFs per GC (not 2) the difference of the best van Rossum error was 15% between models with and without gradients consisting of 100 GC (17% with 2 MF per GC). With a membrane resistance of the PC of 100 MΩ (not 15 MΩ) the difference of the best van Rossum error was 23% between models with and without gradients consisting of 100 GCs

(17% with 15 MΩ). Finally, changing the target sequence to 80, 0, and 120 Hz (not 80, 40, and 120 Hz) resulted in very similar results as obtained with the original target (compare *Figure 8C–E* with *Figure 8—figure supplement 1H–J* and *Figure 8—figure supplement 1D–F* with *Figure 8—figure supplement 1K–M*).

### Code

The Matlab scripts used to reproduce the model results in *Figure 8* are available at: https://github.com/HallermannLab/2019_GC_heterogen (*Straub, 2019*; copy archived at https://github.com/elifesciences-publications/2019_GC_heterogen/settings).

### Statistical testing

Data are expressed as mean ± SEM or as box plots with median and interquartile range. The number of analyzed cells is indicated in the figures. To test for statistically significant differences, we performed Kruskal-Wallis (for three groups) or Mann-Whitney $U$ tests (for two groups) and provide the p values ($P_{Kr-Wa}$, or $P_{Mann-Whitney}$) above the bar-graphs. In case of three groups, we performed non-parametric Dunn's multiple comparisons post-hoc tests and provide the p values in the figure legends ($P_{Dunn}$). Results were considered statistically significant if $p < 0.05$.

## Acknowledgements

We would like to thank Jens Eilers for helpful discussions and for critically reading the manuscript.

## Additional information

### Funding

No external funding was received for this work.

### Author contributions

Isabelle Straub, Conceptualization, Data curation, Software, Supervision, Investigation, Visualization, Methodology, Writing - original draft, Writing - review and editing; Laurens Witter, Data curation, Software, Investigation, Visualization, Methodology, Writing - original draft, Writing - review and editing; Abdelmoneim Eshra, Data curation, Software, Investigation, Visualization, Methodology, Writing - review and editing; Miriam Hoidis, Niklas Byczkowicz, Data curation, Investigation, Visualization; Sebastian Maas, Data curation, Software, Investigation; Igor Delvendahl, Data curation, Software, Writing - review and editing; Kevin Dorgans, Data curation, Software, Investigation, Writing - review and editing; Elise Savier, Martin Krueger, Data curation, Investigation, Writing - review and editing; Ingo Bechmann, Philippe Isope, Data curation, Supervision, Investigation, Writing - review and editing; Stefan Hallermann, Conceptualization, Software, Supervision, Visualization, Methodology, Writing - original draft, Writing - review and editing

### Author ORCIDs

Isabelle Straub (iD) https://orcid.org/0000-0003-1152-5542
Laurens Witter (iD) https://orcid.org/0000-0003-2357-0578
Niklas Byczkowicz (iD) http://orcid.org/0000-0002-6517-287X
Igor Delvendahl (iD) https://orcid.org/0000-0002-6151-2363
Kevin Dorgans (iD) http://orcid.org/0000-0003-1724-6384
Elise Savier (iD) https://orcid.org/0000-0001-7512-1630
Stefan Hallermann (iD) https://orcid.org/0000-0001-9376-7048

### Ethics

Animal experimentation: All experiments were approved in advance by the Institutional Ethics Committees and animals were treated in accordance with the European (EU Directive 2010/63/EU, Annex IV for animal experiments), National, and Leipzig University guidelines.

Decision letter and Author response
Decision letter https://doi.org/10.7554/eLife.51771.sa1
Author response https://doi.org/10.7554/eLife.51771.sa2

## Additional files

### Supplementary files
• Transparent reporting form

### Data availability
All data generated or analysed during this study are included in the manuscript and supporting files. The Matlab scripts used to reproduce the model results in Figure 8 is available at: https://github.com/HallermannLab/2019_GC_heterogen (copy archived at https://github.com/elifesciences-publications/2019_GC_heterogen).

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
