## [Decision Letter]

**Acceptance summary:**

The granule cells within the cerebellar cortex are components of one of the best studied circuits within the vertebrate brain and performs critical computations for motor control and perhaps other functions. This paper reveals an additional axis of complexity across the depth of the granule cell layer, showing that deeper and more superficial neurons are tuned to higher and lower input frequencies respectively. The paper convincingly demonstrates the biophysical basis for this in terms of intrinsic properties like time constant, threshold and input resistance and in terms of the underlying voltage dependent currents. These differences are also matched by anatomical differences in axons that manifest as differences in conduction and in the location and hence dendritic filtering of the output synapses. Together these results alter the way we think about how cerebellar neurons transform their inputs during computations related to behavior. Simulations provide a useful check on these ideas and a quantification of the relative importance of the contributing mechanisms.

**Decision letter after peer review:**

Thank you for submitting your article "Gradients in the cerebellar cortex enable Fourier-like transformation and improve storing capacity" for consideration by *eLife*. Your article has been reviewed by three peer reviewers, including Sacha B Nelson as the Reviewing Editor and Reviewer #1, and the evaluation has been overseen by Barbara Shinn-Cunningham as the Senior Editor.

The reviewers have discussed the reviews with one another and the Reviewing Editor has drafted this decision to help you prepare a revised submission.

Essential revisions:

Two of the reviewers found the treatment of the concept of "gradients" not adequately explained or justified. Reviewer 3 felt that other organizations such as a "salt-and-pepper" organization were not adequately ruled out and felt the case could have been made stronger by including more precise (dare I say "granular?) location information about recorded cells in the various physiology figures and not simply in the anatomical figure. Another reviewer felt that it was unclear whether the authors were arguing for distinct and independent frequency representations, or a more continuous organization. The reviewers and editors felt like these issues could be addressed through a combination of including more details, if available, and textual changes making the proposed organization as well as alternatives not fully ruled out more explicit.

For clarity, reviewer 3's suggestion is: "in Figure 5C, the authors use the granule cell distance from the PCL as independent variable. This is very nice. The reviewer would like to see, besides or instead of the graphs organized in three bands (outer, mid, inner), the graphs organized in scatter plots where the granule cell distance (normalized) is the x-axis. The r-values along with p should also be computed. Besides helping in making the point, this would also help visualize what happens at the level of single data points, which could be masked by taking mean and std error only.

At least the following panels should be integrated with the suggested depiction:

1) Figure 1E, F, H, K, L

2) Figure 2B, C

3) Figure 3D

4) Figure 7C, F

If the authors think that a nonlinear distribution along the layer is present, they could try some simple nonlinearity. "

Title: It is *eLife* policy to include some indication of the preparation/species used. Perhaps it would be appropriate to add "mammalian" to the title.

It is generally *eLife*'s policy that data and code should be made publicly available – usually on a publicly accessible repository.

---

## [Author Response]

Essential revisions:Two of the reviewers found the treatment of the concept of "gradients" not adequately explained or justified. Reviewer 3 felt that other organizations such as a "salt-and-pepper" organization were not adequately ruled out and felt the case could have been made stronger by including more precise (dare I say "granular?) location information about recorded cells in the various physiology figures and not simply in the anatomical figure. Another reviewer felt that it was unclear whether the authors were arguing for distinct and independent frequency representations, or a more continuous organization. The reviewers and editors felt like these issues could be addressed through a combination of including more details, if available, and textual changes making the proposed organization as well as alternatives not fully ruled out more explicit.

We thank the reviews for these suggestions. Unfortunately, we did not measure the exact position of the granule cell within the three investigated zones of the granule cell layer. Therefore, we cannot provide more “granule” location information for the electrophysiological data. However, we took this concern very serious and added five supplementary figures displaying the individual data points and box plots for the respective parameters of the three investigated zones of the granule cell layer. As suggested by the reviewers and the editors, we now explain more explicitly alternatives, which cannot be ruled out.

“The distribution of the raw data (Figure 1—figure supplement 1) suggests a gradual change in the average cell parameters alone the depth axis of the GC layer, but two populations of neurons (*salt and pepper* distribution), or three populations of neurons (inner-, middle-, and outer-zone) cannot fully be ruled out.”

“The genetic reasons for the here-observed gradients in cerebellar cortex are currently not known. Due to a large variability within each zone, our data cannot rule out a *salt and pepper* distribution of two populations of neurons (Espinosa and Luo, 2008).”

For clarity, reviewer 3's suggestion is: "in Figure 5C, the authors use the granule cell distance from the PCL as independent variable. This is very nice. The reviewer would like to see, besides or instead of the graphs organized in three bands (outer, mid, inner), the graphs organized in scatter plots where the granule cell distance (normalized) is the x-axis. The r-values along with p should also be computed. Besides helping in making the point, this would also help visualize what happens at the level of single data points, which could be masked by taking mean and std error only.At least the following panels should be integrated with the suggested depiction:1) Figure 1E, F, H, K, L2) Figure 2B, C3) Figure 3D4) Figure 7C, FIf the authors think that a nonlinear distribution along the layer is present, they could try some simple nonlinearity. "

Unfortunately, we did not measure the exact position of the granule cell within the three investigated zones of the granule cell layer. We therefore cannot provide the suggested graphs. To nevertheless visualize what happens at the level of single data points, we show the individual data points and box plots for the suggested panels in the newly added supplementary figures.

Title: It is eLife policy to include some indication of the preparation/species used. Perhaps it would be appropriate to add "mammalian" to the title.

We have changed the title accordingly.

It is generally eLife's policy that data and code should be made publicly available –usually on a publicly accessible repository.

We have uploaded the Matlab code:

“The Matlab scripts used to reproduce the model results in Figure 8 will be available at: https://github.com/HallermannLab/2019_GC_heterogen.”